# Multi-Stage Approach Using Convolutional Triplet Network and Ensemble Model for Fault Diagnosis in Oil Plant Rotary Machines

Seungjoo Lee [1], YoungSeok Kim [2], Hyun-Jun Choi [2] and Bongjun Ji [3,*]

[1] Korean Peninsula Infrastructure Special Committee, Korea Institute of Civil Engineering and Building Technology, Goyang 10223, Republic of Korea; sjlee@kict.re.kr

[2] Northern Infrastructure Specialized Team, Korea Institute of Civil Engineering and Building Technology, Goyang 10223, Republic of Korea; kimys@kict.re.kr (Y.K.); hjchoi90@kict.re.kr (H.-J.C.)

[3] Department of Regional Infrastructure Engineering, Kangwon National University, Chuncheon 24341, Republic of Korea

\* Correspondence: bjji@kangwon.ac.kr

**Abstract:** Ensuring the operational safety and reliability of rotary machinery systems, especially in oil plants, has become a focal point in both academic and industry arenas. Specifically, in terms of key rotary machinery components such as shafts, the diagnosis of these systems is paramount for achieving enhanced generalization capabilities in fault diagnosis, encompassing multiple sensor-derived variables with their respective fault patterns. This study introduces a multi-stage approach to generalize capabilities for fault diagnosis that considers multiple sensor-derived variables and their fault patterns. This method combines the Convolutional Triplet Network for feature extraction with an ensemble model for fault classification. Initially, vibration signals are processed to yield the most representative temporal and spatial features. Then, an ensemble approach is used to maximize both diversity and accuracy by balancing the contributions of the individual classifiers. The approach can detect three representative types of shaft faults more accurately than traditional single-stage machine learning models. Comprehensive experiments, detailed within, showcase the method's efficacy in diagnosing rotary machine faults across diverse operational scenarios.

**Keywords:** oil plant rotary machines; fault diagnosis; rotor dynamics; Triplet Network; deep metric learning





## 1. Introduction

Oil and gas plants play a pivotal role in the energy sector, producing fossil fuels like petroleum and gas, as well as synthesizing high-molecular organic compounds used in petroleum products [1]. These plants operate through a series of interconnected equipment to facilitate their production processes [2]. Malfunctions of equipment in manufacturing plants can halt subsequent and preceding operations [3]. This interruption can pose risks to workers and lead to delays in product output and a decline in operational efficiency. Fault diagnosis is an essential requirement to avoid these problems [4,5]. Fault diagnosis is a process to swiftly identify the causes of malfunctions and take appropriate remedial measures [6]. The use of fault diagnosis can promptly identify and solve problems, and thereby ensuring worker safety, minimizing production downtime, and reducing economic losses [7]. In oil and gas plants, the incorporation and continual update of a precise fault diagnosis system is essential to ensure safe and efficient operation. Within these plants, numerous processes and equipment are in operation. Among these, the Recycle Gas Compressor (RGD) is used in the desulfurization process to recirculate $H_2$ (hydrogen) and other gases within the system [8].

The RGC is a rotary machine designed to elevate the pressure of hydrogen gas to send it to the reactor under the necessary operational pressure. Critical fault-prone components

of rotary machines include bearings, shafts, seals, and blades or impellers, among others [9]. Historically, research has primarily focused on diagnosing faults first in bearings, then in other components. Among these, the failure of the shaft is common but has received relatively little attention.

In the domain of gear analysis, [10] introduced an improved B-spline to effectively depict the relationship between AR coefficients and the rotating phase. This was aimed at detecting gear tooth cracks and assessing their severity, especially under random speed variations. Moreover, the modified VICAR (MVICAR) model for planetary gearbox vibration detection presented an efficient method for utilizing the rotating speed [11].

Shaft failure can have various causes, including fatigue failure, wear, torsional failure, corrosion failure, erosion, creep, and bending [12–14], but the causes are not easy to identify and diagnose. Traditional methods use signal-processing techniques to collect vibration data for fault diagnosis [9]; examples include Time Domain Analysis methods, such as Root Mean Square, peak-to-peak, kurtosis, and crest factor, which have been widely used [15–17], and Frequency Domain Analysis techniques like Fourier Transform and Wavelet Transform that have also been extensively utilized [18–20]. However, Traditional time-domain and frequency-domain analysis techniques play a pivotal role in detecting defects and abnormal behaviors. However, it is important to understand these techniques were conceived mainly for simpler scenarios. In contemporary real-world environments, characterized by complex machinery and processes, vibration signals often manifest with pronounced variability. This variability arises from numerous factors such as operational changes, external disturbances, and the wear and tear of machinery. Due to such variability, there exists the potential challenge of mischaracterizing genuine defect signals as ambient noise or associating them with benign factors. Consequently, these traditional methods can encounter difficulties in pinpointing early-stage faults amidst the intricate nuances of vibration signals. This point is supported by multiple studies that have highlighted the limitations of these methods in complex environments [20–22]. While traditional methods have shown efficacy, they often encounter challenges with non-linear or anomalous signals, exhibit vulnerability to noise, and may struggle to synthesize insights from both time and frequency domains [23]. Given these constraints, researchers are now exploring methods that leverage machine learning to overcome these limitations.

Machine learning approaches, particularly deep learning models, present a promising alternative to traditional signal-processing techniques [24–27]. By using intricate architectures to analyze vast amounts of data, these models can automatically extract salient features without extensive domain-specific preprocessing. Machine learning-based methods can adaptively recognize intricate patterns and anomalies in the vibration signals [28–30], and thereby significantly increase the accuracy of fault diagnosis.

However, Machine learning models also have their drawbacks. First, when training data are limited or biased toward one or a few outcomes, the model can become overfitted, which means it can describe training data well but cannot describe data that were not used in training [31,32]. Furthermore, deep learning models have an inherent "black box" nature, so the reasons for their decisions may be obscure; this is a significant concern in applications where understanding of the reasoning is important [33,34]. Lastly, deep learning models need a large set of labeled training data, which can be difficult to obtain in practical operation environments.

Deep metric learning has garnered significant interest as a potential solution to these challenges. Deep metric learning can learn meaningful distance metrics between samples [35,36], and therefore may have application in fault diagnosis of RGDs. This ability allows for generalized fault detection without the need for explicit labels for each fault type and can thereby effectively mitigate the overfitting problem. As an example of using metric learning, a semi-supervised method employing adversarial learning and metric learning with limited annotated data was proposed for fault diagnosis of wind turbines. [37]. Moreover, DML can increase the robustness of models and increase the interpretability of their

diagnostic decisions [38]. Therefore, the use of deep metric learning may increase the efficiency, accuracy, and understandability of fault diagnoses in rotary machines.

The intent of this study was to propose and validate a multi-stage approach integrating deep metric learning and ensemble learning to achieve effective and highly accurate diagnosis of shaft faults in RGDs. Understanding the complexities of the shaft and its susceptibility to faults is crucial in the field of rotary machines. We provide three main contributions.

1.  We propose a multi-stage methodology for shaft fault diagnosis, combining the strengths of deep metric learning and ensemble learning. This synergistic approach leverages the capabilities of machine learning to enhance pattern recognition and anomaly detection. Furthermore, it effectively identifies intra-class similarities, using them to differentiate between various pattern classes
2.  To enhance diagnostic efficacy, we employ the triplet loss function, which is designed to reduce intra-class variances and accentuate differences between fault types. This approach ensures our diagnostic model is attuned to subtle shaft anomalies.
3.  Our approach is more accurate than various established machine learning methods in diagnosing diverse types of shaft faults.

This paper is divided into five sections. Section 2 summarizes existing knowledge on this topic. Section 3 describes methods proposed in this study for fault diagnosis. Section 4 presents Results. Section 5 concludes our work and suggests some future research directions.

## 2. Related Work

### 2.1. Use of Using Vibration Signals to Diagnose Faults in Rotary Machines

Fault diagnosis plays a pivotal role in ensuring the smooth operation of industrial and manufacturing systems [39]), especially in the context of rotary machine [40]. A rotary machine encompasses systems wherein components revolve around an axis to generate mechanical energy. These machines are fundamentally composed of essential components such as bearings, stators, gears, rotors, and shafts [41,42], catering to a variety of applications. These machines are integral to functions such as fluid pumping, energy generation in turbines and generators, and operations of fans and compressors [43–45]. A comprehensive review of the existing literature indicates a discernible bias in research emphasis [9]. Conventional studies primarily focus on bearing faults, with rotor and gear faults also receiving significant attention. Despite the critical role of the shaft, research pertaining to shaft faults remains sparse. Furthermore, many of these studies narrowly focus on just one or two types of shaft faults, underscoring a potential research gap.

To diagnose these faults, researchers have turned to a variety of data sources, encompassing acoustic [46–48], thermal [49,50], current [51,52], pressure [53,54], and vibration measurements. Among this spectrum of diagnostic data, vibration analysis has become the main method for predictive maintenance of shaft faults. It can be used to troubleshoot instantaneous malfunctions and guide periodic maintenance. Vibration measurements are typically captured online. They offer real-time diagnostic insights into the machinery's health. Vibrational data, often merged with other parameters, increase the diagnostic interpretation and overall understanding of machine performance.

The subsequent post-data acquisition step involves feature extraction. Methods for this process range from statistical feature extraction techniques like Principal Component Analysis (PCA) to time-frequency representation techniques [55] such as Fourier Transform, Wavelet Transform, and Empirical Mode Decomposition [56]. However, these methods have drawbacks.

A significant challenge is the manual selection of appropriate model parameters for analyzing vibration signals. As data volumes grow and feature dimensions expand, manually selecting model parameters becomes both time-consuming and error-prone. Traditional diagnostic methods would classify machinery as healthy or unhealthy based on whether specific values lie within predefined ranges. However, this basic approach of using

static limit measurements raises questions about its reliability, particularly for intricate machinery. Machine learning techniques use computational power and to identify patterns, so machine learning-driven fault diagnosis methods have been considered a promising tool for the diagnosis of rotating machinery.

### 2.2. Review of Interpretation Methods

Vibration data mostly appears in a time series format, and there are various methods that can be used to analyze this data. The AR (Autoregressive) model and the Varying Index Autoregression (VIA) model are among the commonly utilized methods in time series analysis. However, since these models inherently possess linear characteristics, they have limitations in fully capturing the complex dynamical features of vibration data with nonlinear attributes.

The LSTM (Long Short-Term Memory) model is one of the notable methods for time series data analysis. However, there are specific challenges when detecting anomalies in vibration data. Insufficient data focused on normal vibration patterns increases the risk of the model overfitting. Furthermore, the LSTM model can be highly sensitive to noise and outliers, necessitating the consideration of additional approaches or preprocessing techniques to address these issues.

Machine learning methods like Adversarial Discriminative Learning are primarily used for learning data distributions and generating or transforming new data based on those distributions. However, since the main objective of vibration data analysis is to detect specific trends or states in the data, this method may have limited direct applicability. Considering the characteristics of such models, there is a need for a comprehensive evaluation of the features and limitations of various methodologies to select the optimal machine learning approach for vibration data analysis.

In this study, we aim to enhance the analysis efficiency of vibration data using modern deep learning-based approaches. We extract features of the vibration data using the Convolutional Triplet Network and then build an ensemble model to perform the final prediction. Through a multi-stage approach, we aim to deeply understand the complex characteristics of vibration data and derive more accurate analysis results.

### 2.3. Deep Metric Learning

Deep metric learning is a specialized branch of deep learning that has the goal of detecting and learning similarity metrics from data [57]. The Triplet Network incorporates the foundational principles of deep metric learning [58,59]. It exploits the concept of 'triplets', which are composed of three integral components (Figure 1): an anchor, a positive sample from the same category as the anchor, and a negative sample from a different category.

The formulation ensures the anchor and positive samples represent similar characteristics, whereas the negative sample differs from them distinctly. The Triplet Network can be represented as

$$TripletNet(x, x^{neg}, x^{pos}) = \begin{bmatrix} ||Net(x) - Net(x^{neg})||_2 \\ ||Net(x) - Net(x^{pos})||_2 \end{bmatrix}, \qquad (1)$$

where $x$ is the anchor sample, $x^{neg}$ is a negative sample distinct from the anchor, and $x^{pos}$ is a positive sample * sharing the same class as the anchor. The term $Net(*)$ signifies the embedding of input sample '*' ($\epsilon\{x, x^{neg}, x_{pos}\}$). $||Net(x) - Net(x^*)||_2$ denotes the Euclidean distance between the embeddings of '*' and the anchor sample; i.e., the dissimilarity between the anchor and the negative or positive sample in the embedded space. The anchor and the positive sample both belong to the same category, so $||Net(x) - Net(x^{pos})||_2$ ideally should be small. The objective of the Triplet Network is to ensure in the embedded space, the anchor is closer to the positive sample than to the negative one, typically by a certain margin. This distinction is honed during training by narrowing the difference between these distances.

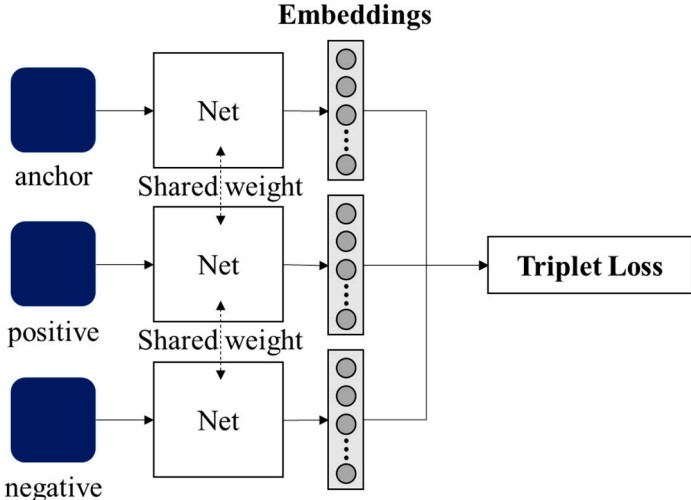

**Figure 1.** Structure of the Triplet Network.

The triplet loss function, a cornerstone of this methodology, is designed with a precise goal: to ensure the distance between the anchor and the positive remains less than the distance between the anchor and the negative, by a stipulated margin. This criterion ensures cohesiveness of embeddings from the same category, while setting those from different categories distinctly apart. The overarching goal is to decrease intra-class variations and heighten inter-class distinctions, thereby crystallizing class boundaries in the embedding space.

$$Loss(d_{pos}, d_{neg}) = \|d_{pos}, d_{neg} - 1\|_2^2 = const \cdot d_{pos}^2 \qquad (2)$$

where

$$d_{pos} = \frac{e^{\|Net(x) - Net(x^{pos})\|_2}}{e^{\|Net(x) - Net(x^{pos})\|_2} + e^{\|Net(x) - Net(x^{neg})\|_2}}, \qquad (3)$$

and

$$d_{neg} = \frac{e^{\|Net(x) - Net(x^{neg})\|_2}}{e^{\|Net(x) - Net(x^{pos})\|_2} + e^{\|Net(x) - Net(x^{neg})\|_2}}, \qquad (4)$$

is designed to ensure that the $d_{pos} < d_{neg}$ between the anchor and the negative, by a stipulated margin. This criterion ensures embeddings from the same category are close to each other, whereas those from different categories are far apart. The goal is to decrease intra-class variations and heighten inter-class distinctions, and thereby crystallize class boundaries in the embedding space.

The neural architecture of the Triplet Network ensures every triplet data point is translated to a concise embedded representation, and is therefore ideal for sequential data processing in fault diagnosis. During successive training iterations, the network uses backpropagation to refine its internal weights, guided by the triplet loss. This iterative refinement persists until the network's loss metrics begin to stabilize; i.e., the model's parameters converge. This optimal stage signifies the network's capability to embed data in a space in which analogous items cluster closely, and disparate ones are far apart.

## 3. The Process of the Multi-Stage Approach

This section outlines the approach used in this study. By ensuring a systematic and replicable approach, we aim to clarify the scientific rigor of our investigation. First, we focus on the generation of relevant data, then describe the processing of generated raw data, then describe advanced feature engineering techniques that use deep metric learning to prepare the data for the final fault diagnosis modeling. Each subsection describes specific methods, tools, and techniques employed in the stages of the research (Figure 2).

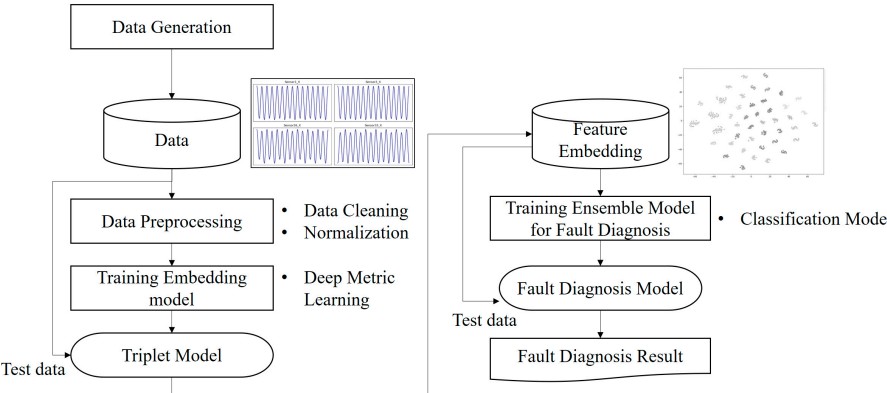

**Figure 2.** Proposed multi-stage approach using convolutional Triplet Network and ensemble model for fault diagnosis in rotary machines.

### 3.1. Data Generation

This study developed a model to describe the operation of the compressor for the desulfurization process. The model focused on identifying and then modeling the crucial shaft components influenced by different fault locations. The design specifications segregated the model into two primary components: the compressor and the turbine (Figure 3).

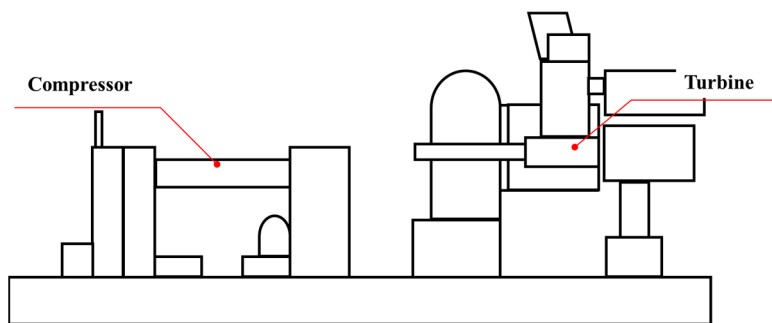

**Figure 3.** Simplified schematic representation of a rotary machine, highlighting the key components: compressor and turbine. The figure has been streamlined to protect proprietary information.

For a realistic scenario, the model was modified to represent the compressors found in the oil plant of a global petroleum and refinery company. The external and internal diameters, and the length of the shaft were specified. The material properties of the shaft were configured as shown in Table 1, after considering various parameters like density, Young's modulus, shear modulus, and Poisson's coefficient, ensuring they are consistent with real-world material properties.

**Table 1.** Shaft Material.

| Properties | Value |
|---|---|
| Density | 7850 kg/m$^3$ |
| Young's Modulus | 217 Gpa |
| Shear Modulus | 81.2 Gpa |
| Poisson Coefficient | 0.299 |

The positions of the sensors, which are critical for the study, were determined (Figure 4) by considering the structure of the compressor and turbine. As referenced in Tables 2 and 3, the rotor discs were described using actual values for mass, polar inertia, and diametral

inertia. For the bearings, the stiffness and damping coefficients were determined according to their actual sizes and positions within the machinery and incorporated into the model.

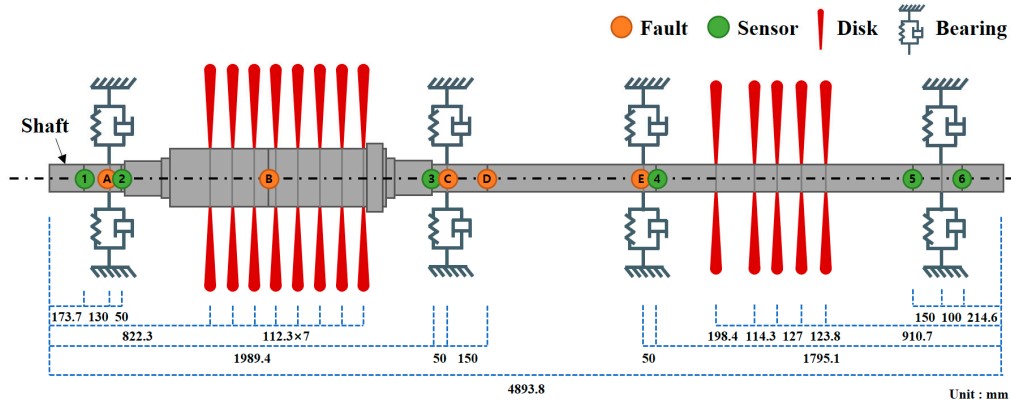

**Figure 4.** Schematic illustration of target rotary machine highlighting the locations of the fault, sensor, and disk within the machinery setup. Orange circles: fault locations; green circles: sensor positions; red cones: disc.

**Table 2.** Disk Properties.

| Properties | Value |
|---|---|
| Mass | 2.6375 kg |
| Moment of Inertia | polar: 0.0075 kg·m$^2$ |
| | diametral: 0.003844 kg·m$^2$ |

**Table 3.** Bearing Properties.

| Properties | Value |
|---|---|
| Stiffness | x-dir: 950 kN/m |
| | y-dir: 109,000 kN/m |
| Damping Coefficient | 50.4 N·s/m |
| | 100.4553 N·s/m |

For the operational scenario, accuracy in the simulation was attained by utilizing the Nyquist theory with a time interval set at 0.0001 s. This was conducted during the rotor dynamics' operational time, which ranged from 0 to 5 s, at a rotational speed of 8400 rpm. It is noteworthy to mention this simulation did not account for the impact of temperature on friction and damping, nor did it consider the effects of inlet/outlet conditions. Consequently, these factors introduce associated sources of uncertainty.

*3.2. Data Preprocessing*

Raw data must be preprocessed to ensure the subsequent analysis is both efficient and provides meaningful results. To extract significant features from vibration data, we used the sliding-window technique as shown in Figure 5. Our dataset was obtained using six distinct sensors (Section 3.1). As a result, for each sensor, the dataset had three columns, one for each axis. Given the intricacies in machinery vibrations and the potential overlapping characteristics across different fault types, the chosen window length must be optimal. The window must be long enough to include meaningful patterns but not so long to introduce irrelevant noise or lose temporal resolution.

| | Time | SensorA_X | SensorA_Y | SensorB_X | SensorB_Y | SensorC_X | SensorC_Y |
|---|---|---|---|---|---|---|---|
| Window 1 | 0.1868 | $6.53 \times 10^{-5}$ | $-3.46 \times 10^{-6}$ | $5.58 \times 10^{-5}$ | $-3.46 \times 10^{-6}$ | $-2.70 \times 10^{-5}$ | $-1.09 \times 10^{-6}$ |
| Window 2 | 0.1869 | $6.51 \times 10^{-5}$ | $-3.17 \times 10^{-6}$ | $5.56 \times 10^{-5}$ | $-3.24 \times 10^{-6}$ | $-2.70 \times 10^{-5}$ | $-1.15 \times 10^{-6}$ |
| | 0.187 | $6.49 \times 10^{-5}$ | $-2.78 \times 10^{-6}$ | $5.55 \times 10^{-5}$ | $-2.95 \times 10^{-6}$ | $-2.70 \times 10^{-5}$ | $-1.26 \times 10^{-6}$ |
| Window 3 | 0.1871 | $6.48 \times 10^{-5}$ | $-2.31 \times 10^{-6}$ | $5.55 \times 10^{-5}$ | $-2.19 \times 10^{-6}$ | $-2.71 \times 10^{-5}$ | $-1.40 \times 10^{-6}$ |
| | 0.1872 | $6.49 \times 10^{-5}$ | $-1.76 \times 10^{-6}$ | $5.55 \times 10^{-5}$ | $-1.75 \times 10^{-6}$ | $-2.71 \times 10^{-5}$ | $-1.57 \times 10^{-6}$ |
| | 0.1873 | $6.49 \times 10^{-5}$ | $-1.14 \times 10^{-6}$ | $5.55 \times 10^{-5}$ | $-1.28 \times 10^{-6}$ | $-2.72 \times 10^{-5}$ | $-1.77 \times 10^{-6}$ |
| | 0.1874 | $6.51 \times 10^{-5}$ | $-4.90 \times 10^{-6}$ | $5.56 \times 10^{-5}$ | $-7.98 \times 10^{-7}$ | $-2.74 \times 10^{-5}$ | $-1.98 \times 10^{-6}$ |
| | … | … | … | … | … | … | … |

**Figure 5.** Schematic representation of the sliding window technique over a data table. Here, the width of the Window is '5'.

### 3.3. Feature Embedding

We used a Triplet Network to transform high-dimensional vibration data to a lower-dimensional representation, to facilitate the extraction of significant features to distinguish various fault conditions from normal conditions (Figure 6).

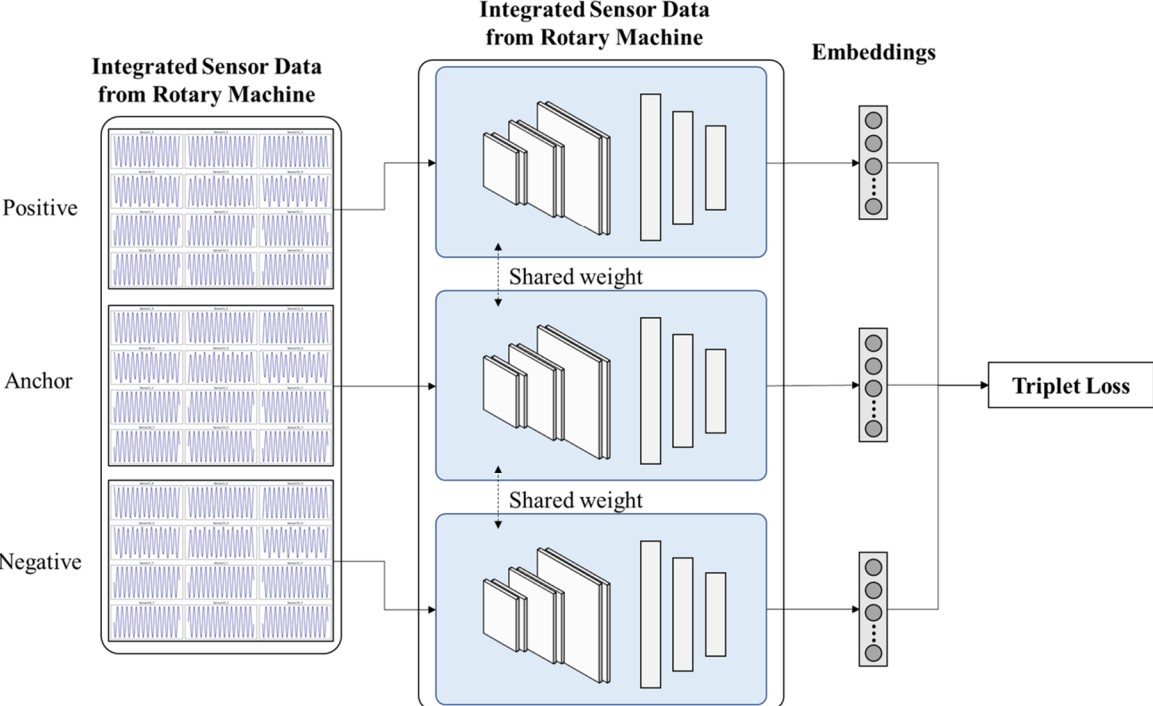

**Figure 6.** Concept illustration of Triplet Network used for feature embedding.

A tailored method to sample triplets was devised to craft an optimal training set for the Triplet Network. This systematic sampling ensures representative exposure to each fault type and location within the training regimen. Our dataset was structured to encompass readings from normal operations and from twelve fault scenarios that represented three fault types each manifested at four locations (Section 3.2).

To exploit the power of the Triplet Network for this dataset, we generated 'triplets' from our data, with the anchor and positive samples being from the same condition, and the negative sample from a different one. To construct these triplets, we selected an anchor sample from a given fault type and location. The positive sample was another instance from the same fault type and location, and thereby ensured intra-class consistency. The negative sample was randomly chosen from any of the other fault types or locations, and thereby guaranteed inter-class diversity.

We fed these constructed triplets into our pre-defined Triplet Network architecture (Section 2). This implementation phase focused on fine-tuning and training the model with our specific dataset. The training was driven by the triplet loss function. Over several epochs, we adjusted the model's weights to minimize the distance between the anchor and positive samples and to concurrently maximize the distance between the anchor and the negative sample in the embedded space. This iterative process continued until the loss values converged, indicating the network had learned optimal embeddings for our data.

The base network (Figure 7) used for the triplet architecture is specifically designed to use 1D convolutional layers to process multiple sensor vibration data. Beginning with the convolutional segment of the network, an initial convolutional layer with 64 filters and a kernel size of 5 is applied, using the Rectified Linear Unit (ReLU) activation function. This choice of activation function is crucial for introducing non-linearity into the model, to enable capture of patterns in the data. The 'same' padding strategy is used to ensure spatial dimensions of the input data are retained after this convolution. The max-pooling operation with a pool size of 2 is applied, to reduce the spatial dimensions while retaining significant features; this process increases computational efficiency. Building on this foundation, the network then uses a second convolutional layer, this time comprising 128 filters, still with a kernel size of 5 and retaining the ReLU activation. 'Same' padding is used again to preserve spatial dimensions and make the architecture predictable. Then, another max-pooling operation with a pool size of 2 is applied to further summarize the data while emphasizing essential features. A third convolutional layer is then deployed; this one has 256 filters and a kernel size of 5, and uses the ReLU activation.

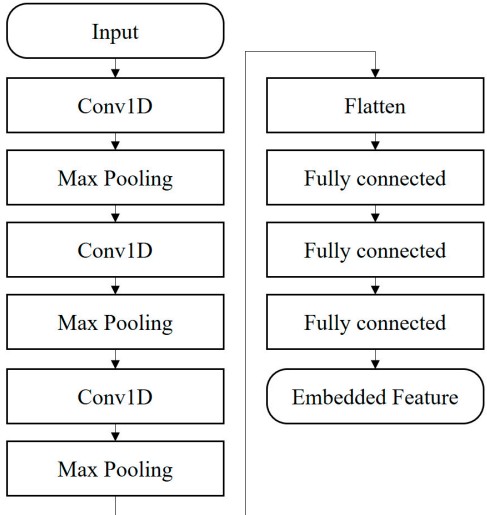

**Figure 7.** Architecture of the base network for employing the Triplet Network.

The increases in filter count from as convolutional layers deepen demonstrates a hierarchical approach, in which each layer captures more intricate and composite features than the previous one. A final max-pooling step with a pool size of 2 is executed, to further encapsulate and simplify the feature map.

During the transition from convolutional layers, the data is subject to a flattening operation that reshapes them to fit the subsequent dense layers. The first step is a dense layer with 256 units that uses the ReLU activation function. The ReLU activation continues to add non-linearity, ensuring the network can model complex relationships. A dropout layer with a rate of 0.2 is interspersed. It randomly deactivates 20% of neurons during training; this process reduces the risk of overfitting. Then another dense layer with 128 units is used, coupled with the ReLU activation. Yet another dropout layer with a rate of 0.2 follows to further guarantee the model's generalizability. Concluding the sequence, a final dense layer transforms the data to the desired embedding space, which by default is set to eight dimensions in the provided configuration.

In essence, this architecture transmutes the vibration data to a compact representation, which is suitable for the demands of the Triplet Network. The blend of convolutional and dense layers ensures both spatial feature extraction and subsequent transformation to a lower-dimensional, yet informative, embedding space. Periodic validation using unseen data triplets from our dataset ensured the model was not overfitting and was generalizing well to new data instances. Upon final training, the Triplet Network effectively mapped the eighteen-dimensional vibration data to an eight-dimensional space, to facilitate clear distinction between normal operational state and various fault conditions.

### 3.4. Fault Diagnosis

To assess the performance of our proposed model, we used accuracy rate $R_A$ as our primary criterion. It measures the ratio of correct predictions to the total number of predictions. The choice to use $R_A$ as an evaluation metric is motivated by its clear interpretability and the critical importance of achieving a high proportion of correct predictions in fault diagnosis.

To further increase the prediction capabilities, we exploit the power of ensemble models, which are known for their ability to combine individual model predictions to boost overall $R_A$. The Random Forest algorithm is an ensemble of decision trees that aggregates the predictions of individual trees to produce a final decision. The Gradient Boosting ensemble model is a sequential boosting algorithm that fits new trees to the residual errors of the preceding ones. The configuration of this model will be shaped by parameters such as learning rate, number of boosting stages, and tree depth. The Voting Classifier acts as a sophisticated ensemble technique that brings together the predictions from multiple models to make a final prediction, typically obtained by majority voting for classification tasks. Within this classifier, predictions can be consolidated by using "hard" or "soft" voting. Hard voting accepts the decision of the majority class predicted by the individual models, whereas soft voting averages the prediction probabilities, and selects the class that has the highest probability. The models that constitute the Voting Classifier, along with any tunable parameters specific to this setup, will also be of interest.

The analysis of these ensemble models used $R_A$ as the comparison criterion. The ensemble model that achieves the highest $R_A$ will be judged to have the highest ability to best capture the intricacies of our dataset and will be chosen as the best for the fault diagnosis of rotary machines.

## 4. Experiment and Result

### 4.1. Data Generation

Using Rotor dynamics Open Source Software [60], we simulated x, y displacement values (Figure 8) at 0.0001-s intervals for each sensor (Table 4). Sensors were placed at six distributed locations, with faults being introduced at five varied locations. The displacement in millimeter unit is collected. This modeling and simulation approach provides detailed understanding of the fault dynamics and their effects, which is crucial for refining operational efficiencies and fault predictions in real-world scenarios.

**Table 4.** Example of generated data of sensor data in rotary machine, the measured values are displacement in millimeter unit.

| Time | SensorA_X | SensorA_Y | SensorB_X | SensorB_Y | SensorC_X | SensorC_Y |
|---|---|---|---|---|---|---|
| 0.1868 | $6.53 \times 10^{-5}$ | $-3.46 \times 10^{-6}$ | $5.58 \times 10^{-5}$ | $-3.46 \times 10^{-6}$ | $-2.70 \times 10^{-5}$ | $-1.09 \times 10^{-6}$ |
| 0.1869 | $6.51 \times 10^{-5}$ | $-3.17 \times 10^{-6}$ | $5.56 \times 10^{-5}$ | $-3.24 \times 10^{-6}$ | $-2.70 \times 10^{-5}$ | $-1.15 \times 10^{-6}$ |
| 0.1870 | $6.49 \times 10^{-5}$ | $-2.78 \times 10^{-6}$ | $5.55 \times 10^{-5}$ | $-2.95 \times 10^{-6}$ | $-2.70 \times 10^{-5}$ | $-1.26 \times 10^{-6}$ |
| 0.1871 | $6.48 \times 10^{-5}$ | $-2.31 \times 10^{-6}$ | $5.55 \times 10^{-5}$ | $-2.19 \times 10^{-6}$ | $-2.71 \times 10^{-5}$ | $-1.40 \times 10^{-6}$ |
| 0.1872 | $6.49 \times 10^{-5}$ | $-1.76 \times 10^{-6}$ | $5.55 \times 10^{-5}$ | $-1.75 \times 10^{-6}$ | $-2.71 \times 10^{-5}$ | $-1.57 \times 10^{-6}$ |
| 0.1873 | $6.49 \times 10^{-5}$ | $-1.14 \times 10^{-6}$ | $5.55 \times 10^{-5}$ | $-1.28 \times 10^{-6}$ | $-2.72 \times 10^{-5}$ | $-1.77 \times 10^{-6}$ |
| 0.1874 | $6.51 \times 10^{-5}$ | $-4.90 \times 10^{-6}$ | $5.56 \times 10^{-5}$ | $-7.98 \times 10^{-7}$ | $-2.74 \times 10^{-5}$ | $-1.98 \times 10^{-6}$ |
| 0.1875 | $6.54 \times 10^{-5}$ | $-1.88 \times 10^{-6}$ | $5.58 \times 10^{-5}$ | $-3.13 \times 10^{-7}$ | $-2.76 \times 10^{-5}$ | $-2.21 \times 10^{-6}$ |

Our dataset consists of normal operational readings and twelve fault scenarios. These consist of three fault types, each in four distinct locations. These twelve fault scenarios are presented in Table 5.

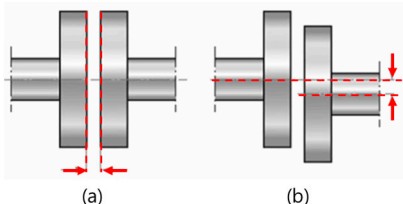

**Figure 8.** Illustration of x (**a**) and y (**b**) value of collected sensor data.

**Table 5.** 12 different fault scenarios.

| Fault Type | Related Parameters |
|------------|-------------------|
| Parallel misalignment | 0.5 mm, 0.1mm, 0.15 mm, 0.2 mm |
| Angular misalignment | 1.25°, 2.5°, 3.75°, 5° |
| Unbalance misalignment | 0.000005 kg·m, 0.00001 kg·m, 0.000015 kg·m, 0.00002 kg·m |

The first fault type is angular misalignment (Figure 9a). It occurs when the shaft's central axis forms a non-zero angle as a result of faulty bearing support. Vibrations due to angular misalignment are primarily axial and have high amplitude. They consist of two coupled components, which are 180° out of phase.

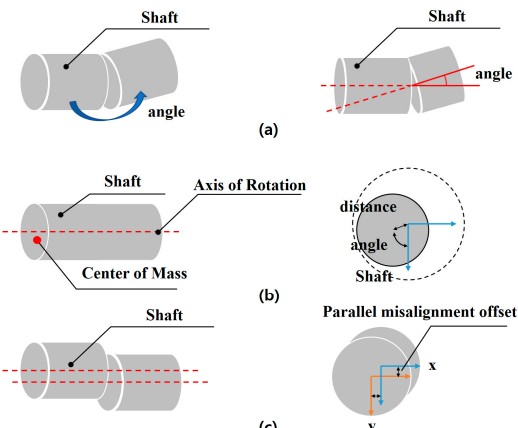

**Figure 9.** Illustrative representation of three fault types considered in this study: (**a**) Angular Misalignment, (**b**) Unbalance, and (**c**) Parallel Misalignment.

The second fault type is unbalance (Figure 9b). It occurs when the center of mass does not coincide with the rotation center. This misalignment results in a centrifugal force, which causes high-amplitude vibrations that have a sinusoidal waveform, typically at the same frequency as the rotation. The amplitude of vibrations due to unbalance increases proportionally to the square of the rotation speed. In rigidly attached machines, the vibration amplitude is greater in the horizontal direction than in the vertical direction. A distinctive characteristic is the 90° phase difference between the horizontal and vertical amplitudes.

The third fault type is parallel misalignment (Figure 9c). It arises when the central axis of the rotating shaft does not align with the line connecting the components that secure it, such as bearings. Such misalignment typically induces substantial vibrations in both radial and axial directions. Vibrations that result from misalignment predominantly have frequencies equivalent to the rotation.

### 4.2. Data Preprocessing

For data preprocessing, a window width of 100 data points (0.01 s) was used to capture short-duration fluctuations and transient characteristics inherent in the vibration signals. To optimize data coverage and to extract overlapping features, a step size of 70 was implemented for the sliding window technique, so consecutive windows overlapped by 30 points. This overlap ensured adequate representation of transitional phases and intermittent patterns that could occur between windows and thereby offered a nuanced understanding of system dynamics.

### 4.3. Feature Embedding

For feature embedding, a comparative assessment was executed using three methods: an Autoencoder [61], PCA, and the Triplet Network. The primary objective was to identify the approach that provides the most meaningful and discernible representation of the vibration data, particularly in distinguishing normal operational conditions from varying fault types.

Once the feature embedded, the t-Distributed Stochastic Neighbor Embedding (t-SNE) technique [62], a nonlinear dimensionality reduction tool, was employed to visualize the embedded results in two dimensions. In the provided labels, the portion of the label preceding the underscore indicates the type of fault. For instance, "angular" referred to an angular misalignment fault, "parallel" denoted a parallel misalignment type of fault, and "unbalance" signified an unbalance fault. On the other hand, the numerical value following the underscore pointed to the location of the fault. As an example, in the label "angular_A", "angular" described the fault type and "A" specified the fault was located at position A. Similarly, "parallel_C" indicated a parallel type fault at the C location, while "unbalance_B" represented an unbalance fault at the B position

This visualization provided an insightful perspective on the clustering and separation capabilities of each embedding method.

In the Autoencoder outcomes (Figure 10a), the embedded features that corresponded to normal operations overlapped significantly with features that corresponded to fault types. Therefore, operational states could not be readily distinguished from fault states. The boundaries between classes were convoluted; this result indicated the Autoencoder's could not extract salient and differentiating features adequately in this dataset.

The Triplet Network outcomes (Figure 10b) aggregated data samples into a discernible cluster for each class, thereby enabling intuitive identification. The boundary demarcation between different fault types and normal operation was clear; this result indicated the Triplet Network effectively identified the structures and disparities within the data.

PCA provided a representation that was intermediate (Figure 10c) between the Autoencoder and the Triplet Network results.

Overall, the Triplet Network was the most effective tool for embedding this specific dataset. The method captured the variances and clustered the different operational states distinctly. The visualization augmented by t-SNE accentuated these differences and emphasized the merits of its embedding strategy for fault detection and classification tasks by analyzing vibration data.

### 4.4. Fault Diagnosis

To investigate the performance differences among various modeling methods, we compared several combinations of embedding techniques and machine learning algorithms (Table 6).

The initial assessment deployed no embedding techniques. In this case, the Support Vector Machine (SVM) and neural network (NN) both obtained $R_A = 0.07$. This significantly low result accentuates the challenge posed by the complex and perhaps high-dimensional feature space. Without any form of preprocessing or feature transformation, these models failed to discern the subtle patterns in the raw data. The ensemble methods Random Forest and Gradient Boosting both obtained $R_A = 0.37$; this result suggests these methods may

have embedded strategies that can identify patterns in raw data. However, AdaBoost and the Voting Classifier both had $R_A$ = 0.22, so they seem to have unable the detection of patterns in the original dataset.

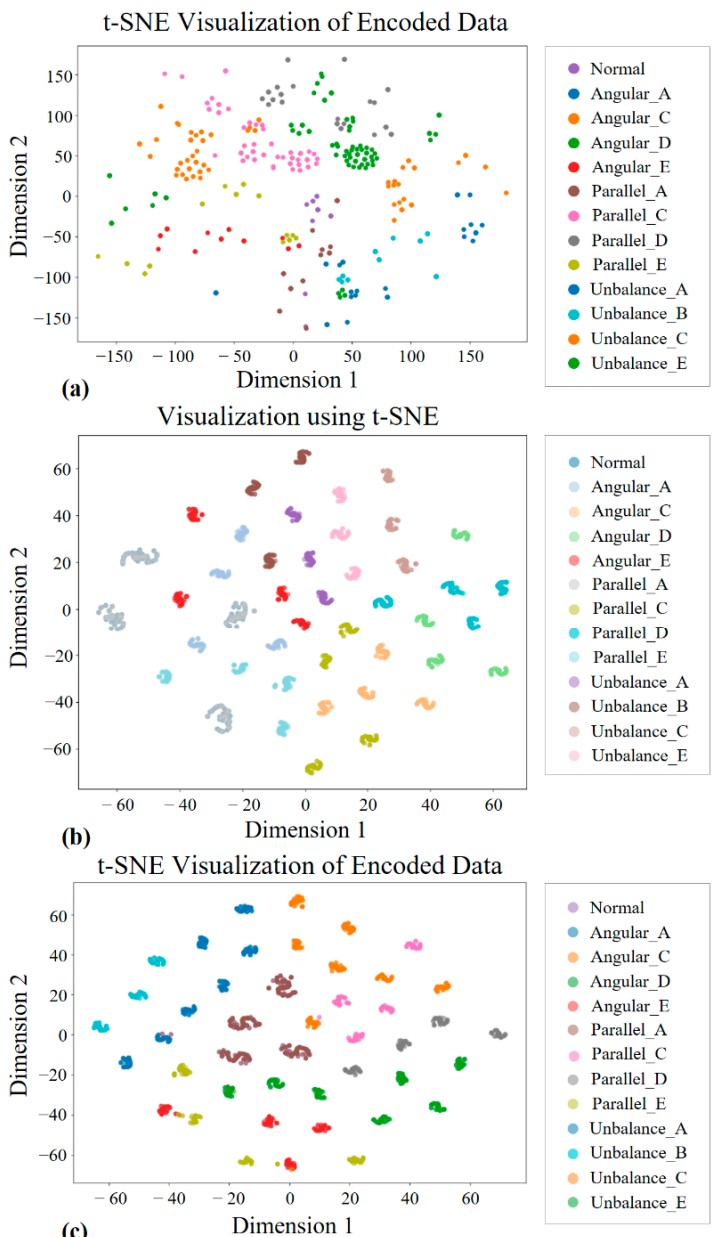

**Figure 10.** Visualization of embedding results using t-SNE for various methods: (**a**) Autoencoder, (**b**) Triplet Network, and (**c**) PCA. In the labels, the text before underscore indicates type of fault: i.e., "angular" = angular misalignment, "parallel" = parallel misalignment, and "unbalance" = unbalance; the number after underscore identifies location of fault; e.g., "angular_A" = angular fault at position A.

After the Autoencoder was used for data embedding, both SVM and NN retained their low $R_A$ = 0.07. This underwhelming consistency across two radically different (i.e., raw vs. autoencoded) data states indicates these methods are not appropriate for this type of fault diagnosis. Gradient Boosting had the highest $R_A$ = 0.45, which suggests it is not adaptable to diverse data representations. Random Forest and AdaBoost has moderate $R_A$ = 0.31 and 0.28 respectively whereas the Voting Classifier and Gradient Boosting had $R_A$ = 0.4 and $R_A$ = 0.45 which still struggles in diagnosing fault.

Use of PCA-embedded data showed an interesting contrast. SVM improved to an impressive $R_A = 0.6$, whereas the NN remained at $R_A = 0.07$. This drastic divergence affirmed SVM's robustness to transformations and indicated the NN may be vulnerable to the dimensional reduction by PCA. The ensemble methods Random Forest, Gradient Boosting, and the Voting Classifier all achieved $R_A = 0.61$; this result indicated PCA was effective in preparing data into a form appropriate for ensemble techniques. AdaBoost, trailed slightly, with $R_A = 0.43$.

**Table 6.** Comparison of accuracy across various embedding methods and ensemble/non-ensemble models.

| Embedding Method | Ensemble | Method | Accuracy |
|---|---|---|---|
| None | Non-ensemble model | SVM | 0.07 |
| | | NN | 0.07 |
| | Ensemble model | Random Forest | 0.37 |
| | | AdaBoost | 0.22 |
| | | Gradient Boosting | 0.37 |
| | | Voting Classifier | 0.22 |
| Autoencoder | Non-ensemble model | SVM | 0.07 |
| | | NN | 0.07 |
| | Ensemble model | Random Forest | 0.31 |
| | | AdaBoost | 0.28 |
| | | Gradient Boosting | 0.45 |
| | | Voting Classifier | 0.40 |
| PCA | Non-ensemble model | SVM | 0.60 |
| | | NN | 0.07 |
| | Ensemble model | Random Forest | 0.61 |
| | | AdaBoost | 0.43 |
| | | Gradient Boosting | 0.61 |
| | | Voting Classifier | 0.61 |
| Proposed (Triplet Network + Ensemble Model) | | | 0.89 |

However, the proposed method achieved an outstanding $R_A = 0.89$. Therefore, this innovative approach set a new benchmark and emphasized the potential benefits of integrating specialized embedding techniques with ensemble models.

To summarize, the traditional models offer varying degrees of success, the incorporation of the Triplet Network distinctly underscores the effectiveness of its feature extraction capabilities. Furthermore, coupling this with ensemble strategies not only underscores a significant advancement in fault diagnosis but also aids in enhancing the model's generalization capabilities across diverse datasets.

## 5. Conclusions

Predictive models for diagnosis of faults in rotary machines must reliably distinguish faulty operation from normal operation and from each other. This paper has reported an evaluation of various combinations of machine learning algorithms and embedding techniques to determine the most effective combination for fault diagnosis. Methods that did not use embedding techniques had notably low accuracy rates $R_A = 0.07$; ensemble models Random Forest and Gradient Boosting had $R_A = 0.37$, AdaBoost had $R_A = 0.22$, and the Voting Classifier, had $R_A = 0.4$; all were unsatisfactory, probably as a result of the complexity and perhaps the high dimensionality of the feature space.

Incorporating the Autoencoder for data embedding did not increase the accuracy of SVM and NN; however, when the ensemble methods Gradient Boosting were applied to the autoencoded data, their $R_A$ increased to 0.45.

The use of PCA as an embedding technique increased the $R_A$ of the SVM model to 0.6; this increase demonstrated remarkable adaptability to linear transformations. In contrast,

$R_A$ of the NN model remained at 0.07. Notably, with the PCA-embedded data, ensemble models Random Forest, Gradient Boosting, and the Voting Classifier, all reached $R_A = 0.61$.

The most significant achievement of our study is our proposed method that consists of a Triplet Network for embedding, integrated with an ensemble model for diagnosis. This combination yields a high $R_A = 0.89$, which confirms the effectiveness of the approach and that merging specialized embedding techniques with ensemble learning methods can increase the accuracy of predictions in complex systems.

In summary, this research demonstrates the need for appropriate selection and integration of embedding and predictive techniques, particularly in complex domains like rotary machine fault diagnosis. The presented multi-stage approach combining the advantages of the Convolutional Triplet Network with ensemble neural networks, is a significant step toward precise and reliable fault diagnosis.

**Author Contributions:** Conceptualization, S.L. and B.J.; investigation: Y.K. and H.-J.C.; formal analysis: S.L., Method: B.J.; Supervision: B.J. All authors have read and agreed to the published version of the manuscript.

**Funding:** This research was supported by a grant (RS-2021-KA161932, Core technology development for AI-based O&M of gas and oil plants) from Ministry of Land Transportation Technology Business Support Program funded by Ministry of Land, Infrastructure and Transport of Korean government. And the authors would like to express their gratitude to Atom Soft (Yeonho Kim) for providing valuable advice.

**Data Availability Statement:** Not applicable.

**Conflicts of Interest:** The authors declare no conflict of interest.

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
