# Peer review of "Multi-Stage Approach Using Convolutional Triplet Network and Ensemble Model for Fault Diagnosis in Oil Plant Rotary Machines"

_machines, doi:10.3390/machines11111012_

Round 1
Reviewer 1 Report
Comments and Suggestions for Authors
This study introduces a multistage approach to generalize capabilities for fault diagnosis that considers multiple sensor-derived variables and their fault patterns. This method combines the Convolutional Triplet Network for feature extraction with an Ensemble model for fault classification. Overall, the paper is well written and organized with a proper length. The contributions as well as the quality are both good. In addition, there are some points that are not very clear and should be addressed in the revised version:
1. The description of the existing work should be shorter in Introduction section and Related Work section. Furthermore, more descriptions of the proposed method are needed.
2. Please update the references especially for recent years since some of the references are too old. For example, multivariate statistical analysis is an important research issue which is widely applied on incipient fault diagnosis of rotary machines. The authors should supplement some results on this aspect.
3. Why the innovative loss function proposed in this paper can ensure that diagnostic model remains sensitive to minute shaft anomalies? Please give some explanations or remarks on this aspect since the reviewers are very interest in incipient fault detection and diagnosis.
Minor editing of English language required
Reviewer 2 Report
Comments and Suggestions for Authors
The authors describe the application of a Multi-stage Approach using Convolutional Triplet Network and Ensemble Model to diagnose the presence of a number of faults in rotary machines employed in oil plants. The paper opens with a description of the study, followed by a brief literature review. The case study is then quickly presented, along with the new methodology.
The new method is then compared against traditional machine-learning alternatives on simulated data set, and the merits of the new solution described.
Although the paper is narrow in scope, it describes an interesting application of triplet networks for fault diagnosis.
The paper is overall well organized and fairly easy to follow, but there are a few significant omissions that need to be addressed before publication.
1) Section 3.1 needs to be expanded and the model described in more detail. It is understood that the case study comes from an industrial application, but it is important to provide a breakdown of the system mass, expected rotating speed, or at least the turbine nominal power. Moreover, it is unclear which non-linearities are accounted for by the model (is friction considered? are non-linear vibration effects considered? etc.)
2) it is unclear which operational scenario is considered during the analysis. Please explain in which condition the system is being simulated and which uncertainty sources have been considered in the study (i.e. the effect of temperature on friction and damping, the effect of different turbine speed or different inlet/outlet conditions).
3) Somewhat related to comments 1 and 2, it is unclear which is the severity of the simulated faults. Please provide the details pertaining the three simulated faults (in example, which is the center of mass eccentricity considered to simulate the rotor unbalance, or which is the angular misalignment etc... provide a table if different values are used). This is integral in understanding if the proposed method is useful for early diagnostics, as hinted in the introductory section.
Without these details it is difficult to understand whether the positive results of the newly proposed method are due to the method itself or to a favorable dataset and make an otherwise well organized paper difficult to recommend.
Comments on the Quality of English Language
No comments.
Reviewer 3 Report
Comments and Suggestions for Authors
1. In the 3rd paragraph of chapter 1 (page 2, line 61) is stated “However, these methods often struggle with the variability of vibration signals, so may not be able to detect faults early”. Please further explain the underlying reason for each of the methods. Such a description is over-judging.
2. In the 6th paragraph of chapter 1 (page 2, line 85) is stated “Deep metric learning has garnered significant interest as a potential a solution to these challenges”. Is here two “a” in “as a potential a solution”?
3. For representation of the temporal features in condition monitoring signals, time series methods like the AR, varying index autoregression model [1], and LSTM [2], are promising tools and should be reviewed and discussed in the lit review section. Additionally, other machine learning methods like adversarial discriminative learning [3] should also be reviewed and discussed.
4. Please provide a more thorough literature review on the topic of fault diagnosis, including recent advances in data-driven methods and their limitations in practical engineering applications. This will help readers understand the significance and novelty of your proposed method. Other relevant methods should be included in the Lit Review section, such as the following papers: doi: 10.1109/TIM.2023.3259048, https://doi.org/10.1016/j.ins.2023.119496, https://doi.org/10.1016/j.ymssp.2020.107605
5. In the 10th paragraph of chapter 1 (page 3, line 108) is stated “Comparison of our approach to various established machine learning methods 108 demonstrate that it is more accurate that they are in diagnosing diverse types of shaft 109 faults.” The expression is a little ambiguous, so whether your way works better, or their way works better?
6. In the 1st paragraph of subsection 2.1 (page 3), this section enumerates too many of the drawbacks of different methods to be grouped into “Related Work”. Please consider whether you would put this part in chapter 1.
7. In Figure 1 of subsection 2.2 (page 4), the picture should be centered, there are other similar problems in the text.
8. In the 4th paragraph of subsection 2.2 (page 4, line 178) is stated “The Triplet Loss function is, a cornerstone of this methodology, is designed with a precise goal:”, please correct the grammar of this sentence.
9. In the 4th paragraph of subsection 2.2 (page 5, line 187), the font of the formula is inconsistent with formula(3) or formula(4). Please make them consistent.
10. In Figure 4 of subsection 3.1 (page 6), the picture is not clear, the words have overlapped. Please draw a new one.
11. In Figure 7 of subsection 3.3 (page 8), the text in the picture is too large, please adjust the font size to make it harmonize.
12. In Table 1 of subsection 4.1 (page 10), there is a problem the table is out of the left bound, please adjust the size of this table to fit the size of the paper. There are other similar problems in the text.
13. In the 2nd paragraph of subsection 4.3 (page 11, line 382) is stated “After feature embedding”, please consider whether such writings fit the theme of this section “Feature embedding”.
14. At the end of page 12, there is a large white space, but you haven’t ended the chapter, please adjust the text to avoid this blank space.
15. In Figure 10 of subsection 4.3 (page 13), we can't see the text clearly in the picture, there are other similar problems in the text.
Comments on the Quality of English Language
Moderate editing of English language required
Round 2
Reviewer 1 Report
Comments and Suggestions for Authors
In a general way most of my comments were answered by the authors. My overall opinion about this paper is quite good. The manuscript is well written and acceptable for publishing
Comments on the Quality of English LanguageMinor editing of English language required
Reviewer 3 Report
Comments and Suggestions for Authors
The authors have presented responses to the comments and made appropriate modifications. It can be accepted in my opinion, I have no further comments.
Comments on the Quality of English LanguageMinor editing of English language required